# Tilapia Lake Virus Does Not Hemagglutinate Avian and Piscine Erythrocytes and NH_4_Cl Does Not Inhibit Viral Replication In Vitro

**DOI:** 10.3390/v11121152

**Published:** 2019-12-12

**Authors:** Augustino Alfred Chengula, Stephen Mutoloki, Øystein Evensen, Hetron Mweemba Munang’andu

**Affiliations:** 1Department of Basic Sciences and Aquatic Medicine, Faculty of Veterinary Medicine, Norwegian University of Life Sciences, P.O. Box 369, NO-0102 Oslo, Norway; achengula@yahoo.com (A.A.C.); stephen.mutoloki@nmbu.no (S.M.); oystein.evensen@nmbu.no (Ø.E.); 2Department of Veterinary Microbiology, Parasitology and Biotechnology, College of Veterinary Medicine and Biomedical Sciences, Sokoine University of Agriculture, P. O. Box 3019 Chuo Kikuu, Morogoro, Tanzania

**Keywords:** red blood cells, Tilapia lake virus, hemagglutination, ammonia chloride, inhibition

## Abstract

Tilapia lake virus (TiLV) is a negative-sense single-stranded RNA (-ssRNA) icosahedral virus classified to be the only member in the family Amnoonviridae. Although TiLV segment-1 shares homology with the influenza C virus PB1 and has four conserved motifs similar to influenza A, B, and C polymerases, it is unknown whether there are other properties shared between TiLV and orthomyxovirus. In the present study, we wanted to determine whether TiLV agglutinated avian and piscine erythrocytes, and whether its replication was inhibited by lysosomotropic agents, such as ammonium chloride (NH_4_Cl), as seen for orthomyxoviruses. Our findings showed that influenza virus strain A/Puerto Rico/8 (PR8) was able to hemagglutinate turkey (*Meleagris gallopavo*), Atlantic salmon (*Salmo salar* L), and Nile tilapia (*Oreochromis niloticus*) red blood cells (RBCs), while infectious salmon anemia virus (ISAV) only agglutinated Atlantic salmon, but not turkey or tilapia, RBCs. In contrast to PR8 and ISAV, TiLV did not agglutinate turkey, Atlantic salmon, or tilapia RBCs. qRT-PCR analysis showed that 30 mM NH_4_Cl, a basic lysosomotropic agent, neither inhibited nor enhanced TiLV replication in E-11 cells. There was no difference in viral quantities in the infected cells with or without NH_4_Cl treatment during virus adsorption or at 1, 2, and 3 h post-infection. Given that hemagglutinin proteins that bind RBCs also serve as ligands that bind host cells during virus entry leading to endocytosis in orthomyxoviruses, the data presented here suggest that TiLV may use mechanisms that are different from orthomyxoviruses for entry and replication in host cells. Therefore, future studies should seek to elucidate the mechanisms used by TiLV for entry into host cells and to determine its mode of replication in infected cells.

## 1. Introduction

Tilapia lake virus (TiLV) causes high mortality rates in wild and farmed tilapia. The disease caused by TiLV was first reported in Israel in 2009, while the virus was first characterized in 2014 [1]. Since then, TiLV infections have been reported from different countries around the world [2,3,4,5]. It is an icosahedral virus made of a trilaminar capsid with a diameter of 55–110 nm, which is morphologically similar to orthomyxoviruses [1]. Its genome is made of -ssRNA consisting of 10 segments, of which segment 1 has a weak homology with the influenza virus C PB1 subunit (~17% amino acid identity, 37% segment coverage). In addition, it has four motifs (I–IV) conserved in RNA-dependent RNA and RNA-dependent DNA polymerases that are homologous with influenza A, B, and C viruses [6]. As such, TiLV was initially proposed to be an orthomyxo-like virus belonging to the genus *Tilapinevirus* based on ultrastructural, chemical, and weak sequence similarities with orthomyxoviruses, but was recently classified as the only genus of the family *Amnoonviridae* of the order Articulavirales. It is transmitted horizontally from infected to susceptible fish [1], although vertical transmission has also been reported [7]. Clinical signs include lethargy, exophthalmia, haemorrhages, irregular swimming, and skin erosions [1]. Mortality ranges between 10% and 90% in TiLV-infected tilapia. Mortality caused by TiLV has been reported in various countries such Egypt, Thailand, Peru, Malaysia, India, and Ecuador [1,2,5,8,9,10,11,12,13]. Current control measures are mostly focused on the implementation of biosecurity measures.

Influenza A and B viruses contain envelope-associated proteins, namely, hemagglutinins (HA) and neuraminidase (NA), inserted into the lipid bilayer, which protrude from the outer surface as spikes. HAs play vital roles in virus entry by binding to sialic acid (Sia) on epithelial cell surfaces, which promotes fusion of the envelope to the cell membrane [14,15]. On the other hand, NA cleaves sialic acid on virion proteins to prevent clumping and facilitates virus release from infected cells [16]. During viral infection or budding, HA is activated when its precursor, HA0, is cleaved by membrane-bound host trypsin-like proteases into HA1 (50 kDa) and HA2 (25 kDa) subunits [15,17,18]. The HA1 subunits of the influenza A and C viruses contain a shallow pocket of conserved amino acids at the distal tip (receptor-binding site) that binds to Sia receptors, either α2-3 or α2-6 linkages [19]. The HA2 subunit contains the hydrophobic peptide required for membrane fusion [15,20]. Sialic acids (Sia) are among the class of monosaccharides containing a nine-carbon backbone, which normally attach to the glycans of animal cells [17,21]. In general, red blood cells (RBCs) from different host species possess both α2-3 and α2-6 linkages, where viruses containing HAs bind [22]. For example, influenza viruses agglutinate avian, mammalian, and piscine RBCs, while infectious salmon anemia virus (ISAV) agglutinate RBCs from cold-water-adapted piscine species [23]. The term hemagglutinin was coined due to the binding ability and aggregation (hemagglutination) of RBCs by the virus [16]. This viral property led to use of the HA assay, first introduced by Hirst [16] in the early 1940s, as a method to identify and quantify viruses that agglutinate RBCs. The HA assay is based on the principle of hemagglutination, in which HA binds to Sia receptors on RBCs, forming lattices that keep RBCs in suspension and appearing as a diffuse suspension. The ability of TiLV to agglutinate RBCs from tilapia or other host species has not been determined, despite the identified similarities with orthomyxoviruses.

Following HA binding to Sia receptors, virus internalization into host cells is mostly dependent on the endocytic pathway, leading to virus localization in endosomal vesicles, where uncoating is facilitated by fusion of the viral envelope with endosomal membranes, further leading to viral cargo delivery into cytoplasm [24]. The fusion of viral envelopes with endosomal membranes is highly dependent on the low endosomal pH (<6.0). Inhibiting endosomal acidification using weak basic lysosomotropic agents such as ammonium chloride (NH_4_Cl) and chloroquine that diffuse across lysosomal membranes due to protonation have been shown to interfere with orthomyxovirus replication [25,26]. Hence, in the present study, we wanted to determine whether the replication of TiLV could be inhibited by endosomal membrane fusion inhibitors, such as NH_4_Cl. Given that TiLV has been shown to have some properties that are similar to influenza viruses, we also wanted to determine whether TiLV agglutinated piscine and avian RBCs, as seen in influenza viruses and ISAV. We found that TiLV did not agglutinate RBCs from any of the species included in the present study, thereby suggesting an absence of surface proteins with hemagglutination properties, pointing toward receptors other than sialic acid types for uptake. Resistance toward lysosomotropic agents indicated that the viral uptake mechanisms were independent of acidification.

## 2. Materials and Methods

### 2.1. Viruses and Cells

E-11 cells (ECACC 01110916, European Collection of Authenticated Cell Cultures, Salisbury, UK) and SHK-1 cells (ECACC 97111106, Salisbury, UK) were cultivated in Leibovitz L-15 medium Glutamax^®^ (Glutamax, Gibco, Gaithersburg, MD, USA) and supplemented with 10% (*v*/*v*) fetal bovine serum (FBS) (Sigma-Aldrich, St. Louis, MO, USA) at 28 °C and 20 °C, respectively, until formation of near confluent monolayers (80–90%). E-11 cells were used to propagate TiLV, while SHK-1 cells were used to grow ISAV. Near confluent monolayers were washed twice with phosphate-buffered saline (PBS), followed by adding the virus suspension and incubation for 1 h at 28 °C (TiLV) and 15 °C (ISAV) to allow virus adsorption onto the cells. The cell culture flasks were gently tilted to ensure uniform distribution of the virus solution on the cell monolayer. After 1 h of virus adsorption, the unbound virus was washed off three times using PBS and freshly prepared growth media containing 2.5% FBS was added to the cells. Cell culture flasks infected with TiLV were cultured at 28 °C, while ISAV-infected flasks were cultured at 15 °C. The virus was harvested when the cytopathic effect (CPE) was fully developed by centrifugation of the supernatant from each flask at 2500 rpm for 10 min. The virus supernatant harvested from each flask was filtered at a size of 22 μm to remove cell debris. Each virus was stored at −80 °C until use. The influenza virus strain A/Puerto Rico/8, designated as PR8 in this study, was grown in the allantois membranes of 11-day-old embryonated chicken eggs and harvested after 3 days. The PR8 virus was kindly provided by Professor Espen Rimstad of Norwegian University of Life Sciences (NMBU) in Norway.

### 2.2. Virus Titrations

Quantification of TiLV and ISAV was done in E-11 and SHK-1 cells cultured in 96-well plates, respectively, using the tissue culture infective dose (TCID_50_/mL) method. The titer for influenza virus strain A/Puerto Rico/8 (PR8) was determined using the HA assay with chicken red blood cells.

### 2.3. Verification of Virus Species Used for the Hemagglutination Assays

Verification of virus species used for the HA assay was carried out using total RNA extracted from cell culture supernatants for the TiLV and ISAV, whereas the allantois fluid was used for the extraction of total RNA for influenza virus PR8 using a modification of the Trizol (GIBCO, Life Technologies, Gaithersburg, MD, USA) and RNAeasy Mini kits (Qiagen, Hilden, Germany), as previously described [27]. cDNA synthesis was carried out in 20 μL reaction volumes using the Qiagen quantiTect^®^ reverse transcription kit that involved an integrated step for the removal of contaminated genomic DNA (Qiagen), as previously described in our studies [28]. TiLV, PR8, and ISAV nucleic acids were amplified using primers specific to each virus, as shown in Table 1, targeting segment 3 for TiLV, segment 7 for PR8, and segment 8 for ISAV. PCR reactions were carried out using the Qiagen kit according to the manufacturer’s guidelines. RNase-free water was used as a negative control.

### 2.4. Collection and Preparation of Erythrocytes

Red blood cells (RBCs) used for the HA assay were collected from turkey (*Meleagris gallopavo*), Nile tilapia (*Oreochromis niloticus*), and Atlantic salmon (*Salmo salar* L.). Whole blood from Nile tilapia and Atlantic salmon cultured at the Norwegian University of Life Sciences aquarium was collected from the caudal vein of each fish and immediately mixed with an equal volume of Alsever’s solution (A3551, Sigma-Aldrich). Fish used for blood collection were anesthetized using benzocaine (20 mg/L). Blood from turkey was obtained from NAISER Company (Oslo, Norway), provided on a commercial basis. The blood from turkey was collected and immediately mixed with an equal volume of Alsever’s solution. Thereafter, whole blood in the Alsever’s solution from turkey, tilapia, and salmon was washed in PBS to remove plasma, thereby leaving RBCs for the HA assay. For washing, 5 mL of RBCs was diluted in 45 mL PBS at room temperature, followed by centrifugation at 2000 rpm for 10 min at 4 °C; the supernatant was discarded. The washing in PBS was repeated three times and the pellet containing the RBCs was estimated using the graduated marks on the tubes after the final washing and centrifugation. This was followed by 10% dilution of RBCs in PBS (1:10) to prepare a stock solution. Finally, the 0.5% RBC working solution for HA test was prepared by diluting the RBC suspension in PBS at a ratio of 1:20, giving a final suspension of 0.5% RBCs from the stock solution. The 0.5% RBC working solution was prepared every day, while the RBC stock solution was only used up to a maximum of three days.

### 2.5. Hemagglutination Assay

Viruses used for the HA assay were TiLV (in four concentrations of 10^7^, 10^6^, 10^5^ and 10^4^ TCID_50_/mL), ISAV (10^6^ TCID_50_/mL), and PR8 (512 HA/50 μL). The HA assays were performed in V-shaped 96-well plates (ThermoFisher Scientific, Waltham, MA, USA), as previously described [31]. Briefly, 50 μL of PBS was added to all wells. Thereafter, 50 μL of virus was added to the first column (column-1), giving a total of 100 μL. After mixing, 50 μL was transferred to wells in the next column (column-2). This two-fold serial dilution of mixing and transferring of 50 μL into the next column was continued until the last column on the plate, at which 50 μL, after mixing in the last column (column-12), was discarded. Thereafter, 50 μL of 0.5% RBC was added to all wells, followed by gentle mixing. The plate was monitored for hemagglutination at room temperature for up to 60 min. Wells showing a uniform reddish color of RBC suspension were regarded as positive for the HA assay, while wells with a red dot in the center at the bottom of each V-shaped well after one hour of incubation at room temperature were regarded as negative. As a measure of non-agglutination (negative results), stripping of unbound RBCs settled at the bottom of each well was performed by tilting the plate at a 60° angle. The highest dilution showing a uniform reddish color of RBCs was regarded as the HA titer of the virus tested.

### 2.6. Evaluating the Effect of Ammonium Chloride on TiLV Replication in E-11 Cells

E-11 cells were seeded into T-25 culture-flasks in L-15 growth media, followed by incubation at 28 °C until 95% confluence. Confluent monolayers were washed twice using PBS before treatment using various concentrations of NH_4_Cl. Experiments using NH_4_Cl treatment of E-11 cells were done in two parts, of which the first part was to determine the toxicity of different concentrations of NH_4_Cl on E-11 cells. To do this, the concentrations of NH_4_Cl tested were 1, 5, 20, 25, 30, and 50 mM. Once the different NH_4_Cl concentrations mixed with growth media were put on the cells followed by incubation at 28 °C, the cells were observed every hour to check for toxicity based on morphological changes and cell death. The second part was to test the impact of NH_4_Cl on replication of TiLV in E-11 cells once the optimal duration that did not cause toxicity on the cells was determined. To do this, cells were treated with NH_4_Cl during virus adsorption, while post-adsorption treatment was done at 1, 2, or 3 h post-infection (hpi). Cells were infected with TiLV at multiplicity of infection (MOI) 1.0 and adsorbed for 1 h at 28 °C to allow for virus binding onto E-11 cells. The cell culture flasks were gently tilted to ensure uniform distribution of the virus solution on the cell monolayer. After 1 h of virus adsorption, any unbound virus was washed off three times using PBS. The cells were cultured in normal growth media containing 5% FBS after NH_4_Cl treatment. Three negative control cell culture flasks without NH_4_Cl treatment and cells treated with NH_4_Cl but without virus infection were included in the study. The positive control group consisted of cells infected with the virus at MOI 1.0 without NH_4_Cl treatment. All cells were incubated at 28 °C and monitored daily for cytopathic effect (CPE) development. Once CPE was observed, total RNA was extracted for virus quantification from all treatment groups using a modification of the Trizol (GIBCO, Life Technologies) and RNAeasy Mini kit (Qiagen) methods, as previously described [32,33]. Quantitative real time PCR (qRT-PCR) was performed on triplicates of the uninfected and infected cells using primers targeting TiLV segment 3 (Table 1).

## 3. Results

### 3.1. Verification of the Viruses Used in the Hemagglutination Assay

Species-specific primers were used to detect the different viruses used for the hemagglutination assay by PCR followed by gel electrophoresis, and the amplified products were specific for each of the viruses (Appendix A). Detection of the 109 bp band for TiLV (lane TiLV), the 195 bp band for PR8 (lane PR8), and the 104 bp band for ISAV (lane ISAV), as shown in Appendix A, was in concordance with the expected amplicon products for each virus, as shown in Table 1. Note that no bands were detected in the RNase free water in the NC lanes tested against TiLV, PR8, and ISAV primers, thereby indicating that there was no unspecific binding for the primers used in the gel electrophoresis analysis.

### 3.2. TiLV Hemagglutination Test Using Avian Erythrocytes

Red blood cells from turkey were used as representative of avian species for the HA assay. The viruses used in the assay were PR8 titrated in a two-fold dilution in row A, starting with the highest concentration of 512 HA units, while ISAV was titrated in a two-fold dilution in rows B and C, starting with the highest concentration of 10^6^ TCID_50_/mL in column 1. The HA titer used for PR8 had no direct link to the TiLV and ISAV TCID_50_/mL. As shown in Figure 1, four concentrations of TiLV, starting with 10^7^, 10^6^, 10^5^, and 10^4^ TCID_50_/mL, were diluted two-fold in rows D, E, F, and G, respectively, while row H was used as a negative PBS control. Note that the highest dilution factor at which PR8 agglutinated turkey erythrocytes was 1:128/50 μL, as shown in Figure 1, while the weak stripping of unbound turkey RBCs for the same plate were held at a 60° angle, as shown in Appendix A. Elution of the PR8 from turkey erythrocytes was not observed even after 4 h of incubation at room temperature and 12 h incubation at 4 °C. However, there was no agglutination of turkey erythrocytes by ISAV and TiLV (Figure 2) and no stripping of RBCs, as shown in Appendix A, when the same plate was held at a 60° angle in order to detect the stripping of non-agglutinated RBCs, similar to observations seen in the PBS negative control in row H. In summary, only PR8 agglutinated turkey erythrocytes, while neither of the piscine viruses (TiLV and ISAV) agglutinated the turkey RBCs (Table 2).

### 3.3. TiLV Hemagglutination Test Using Piscine Erythrocytes

Tilapia and Atlantic salmon RBCs were used as representatives of piscine species for the HA assays. Figure 2 shows the layout of the viruses used in the assay, of which TiLV was used starting with a concentration of 10^7^ TCID_50_/mL diluted two-fold in rows A and B, PR8 (512 HA/50 µL) in rows C and D, and ISAV (10^6^ TCID_50_/mL) in rows E and F, while rows G and H contained the PBS negative control. Figure 2 shows that there was no agglutination of tilapia erythrocytes by TiLV, as demonstrated by rows A and B being similar to the PBS negative control in rows G and H. Rows E and F in Figure 2 showed weak agglutination of tilapia RBCs by ISAV at a 1:4 dilution, while Appendix A shows that there was no stripping of RBCs at the 1:4 dilution (rows E and F) on the hemagglutination plate. The stripping became more prominent at the 1:8 dilution onward, which was similar to the PBS negative control. However, rows C and D showed that the highest dilution factor at which PR8 agglutinated tilapia RBCs was 1:64/50 μL.

Rows A and B, as seen in Figure 3 and Appendix A, showed that the highest dilution factor at which ISAV agglutinated Atlantic salmon erythrocytes was 1:16/50 μL, whereas the highest dilution factor for PR8 was 1:512/50 μL. Elution of Atlantic salmon agglutinated RBCs by PR8 and ISAV was not observed after 4 h of incubation at room temperature and 12 h of incubation at 4 °C. Rows E and F (Figure 3) showed that there was no agglutination of Atlantic salmon erythrocytes by TiLV, which was a similar observation as that seen in rows G and H for the PBS negative control.

In summary, these findings showed that PR8 was able to agglutinate turkey, Atlantic salmon, and tilapia RBCs, while ISAV only agglutinated Atlantic salmon and tilapia RBCs. In contrast, TiLV was not able to agglutinate turkey, tilapia, or Atlantic salmon RBCs (Table 2).

### 3.4. Evaluating the Effect of Ammonium Chloride on TiLV Replication in E-11 Cells

Treatment of E-11 cells with different concentrations of NH_4_Cl showed that leaving the NH_4_Cl media for more than 24 h on the cells was toxic for concentrations of ≥50 mM NH_4_Cl. Lower concentrations, i.e., between 1 and 30 mM, had no toxic effect even when left on the cells until the end of the experiments. When the NH_4_Cl media was replaced by maintenance media (L15 plus 2.5% FBS) after 7 h of NH_4_Cl treatment had no toxic effect on the cells for high concentrations (≥50 mM NH_4_Cl). Hence, a concentration of 30 mM was selected to test the effect of NH_4_Cl on TiLV replication, which was replaced with maintenance media after NH_4_Cl treatment. The choice of 30 mM was in the range of NH_4_Cl concentration used in several previous studies [34,35,36]. Therefore, in the pre-treatment group, NH_4_Cl-containing media was added 30 min before virus adsorption, NH_4_Cl treatment was performed during virus adsorption in the group designated as 0 hpi, and NH_4_Cl media was added after 1, 2, and 3 hpi and was later replaced with maintenance media in the post-infection groups. The cytopathic effects (CPE) seen at 3 days post-infection (dpi) included the formation of vacuoles (patches) of detached cells from E-11 monolayers infected with TiLV (Appendix A). Figure 4 shows that there was no significant difference in the mean viral titer based on *C*t-values detected by qRT-PCR from three replicates of TiLV-infected cells treated with 30 mM NH_4_Cl at 0, 1, 2, and 3 hpi. In addition, there was no significant difference in the mean virus titer detected in the positive control (without NH_4_Cl treatment) and the groups treated with NH_4_Cl at 0, 1, 2, and 3 hpi.

## 4. Discussion

The main finding in this study was that TiLV did not agglutinate tilapia or Atlantic salmon RBCs, and that NH_4_Cl treatment neither inhibited nor enhanced TiLV replication in E-11 cells. The influenza virus strain A/Puerto Rico/8 (PR8) agglutinated avian and piscine RBCs, while ISAV only agglutinated Atlantic salmon and tilapia RBCs. The agglutination of tilapia RBCs by ISAV observed in this study was weak. The inability of TiLV to agglutinate RBCs from any of the species included suggests an absence of surface proteins able to agglutinate RBCs. A lack of an observed effect of lysosomotropic agents indicates an acidification-independent uptake mechanism, although additional studies should be carried out to understand the details of the mechanisms involved.

The observed ISAV properties are in line with observations made by Falk et al. [25], showing that ISAV only agglutinated piscine RBCs and not avian RBCs. Similarly, it was shown by the same authors that influenza virus A agglutinated both avian and piscine RBCs, which corroborated our findings.

Different orthomyxoviruses bind to different Sia receptors on host cells. In human influenza A and B viruses, the HA1 subunit binds to Sia receptors attached to either α2-3 or α2-6 linkages [19,21], avian and equine influenza viruses bind Sia containing the α2-3 linkage [17], and swine influenza viruses bind Sia attached to the α2-3 and α2-6 linkages [37,38]. Erythrocytes from several species, such as chicken, turkey, equine, guinea pigs, and humans, possess both α2-3 and α2-6 linkages [22]. Hence, it is likely that the influenza virus strain PR8 used here binds α2-3 and α2-6 linkages on RBCs found in different host species, including avian and piscine species. Paramyxoviruses, such as the Newcastle disease virus (NDV), share the Neu5,9Ac_2_ antigenic determinant with orthomyxoviruses [39]. Other viruses that use the Neu5,9Ac_2_ receptor include coronaviruses. This receptor has been reported in various species, such as humans, rabbits, rats, guinea pigs, horses, chicken, turkey, goose, salmonids, crucian carp, mollusks, and sea urchins [40,41,42,43,44,45,46,47,48,49,50]. Unlike influenza C, ISAV specifically binds Neu4,5Ac_2_ glycans on host cells using its hemagglutinin-esterase (HE) protein. This HE is unique when compared with the HA or hemagglutinin-esterase-fusion (HEF) proteins of other orthomyxoviruses, where the sequence identity is very low (<10%). The ISAV HE sequences shares sequence similarity with non-orthomyxoviruses, such as toroviruses and coronaviruses, by <25% [51,52,53]. It is this high specificity of the ISAV HE protein for Neu4,5Ac_2_ glycans which make it different from other orthomyxoviruses, and may be what accounts for ISAV having lower tropism and mostly binding to fish RBCs, unlike other orthomyxoviruses which bind a wide range of RBCs from different avian, mammalian, and piscine species.

Given that the HA and HE proteins that bind RBCs in hemagglutination assays also serve as ligands that bind Sia receptors used for virus entry into host cells, the inability of TiLV to bind RBCs suggests that it uses mechanisms that are different from orthomyxoviruses for entry into and replication within host cells. This observation is supported by our findings regarding the NH_4_Cl treatment of E-11 cells infected by TiLV. Studies into cell attachment and replication mechanisms of orthomyxoviruses showed that after HA binding to Sia receptors on host cells, the virus is taken up into endosomal compartments by endocytosis. The low endosomal pH mediates the fusion of viral envelopes with endosomal membranes to facilitate virus uncoating, leading to release of nucleoproteins required for virus replication by acid-dependent cellular proteases. The pH dependence of viral uncoating has been completely paralleled with the fusion of viral envelopes with endosomal membranes [35]. As such, raising the endosomal or lysosomal pH by adding weak lysosomotropic bases like chloroquine and NH_4_Cl has been shown to prevent virus uncoating, thereby blocking virus replication [35,54]. This phenomenon is supported by several studies [55,56,57,58,59] including Matlin et al. [60], who showed that NH_4_Cl treatment of Madin–Darby canine kidney (MDCK) cells was able to block the uncoating of the influenza virus in endosomal compartments, leading to replication failure and culminating in the prevention of infection. Other viruses where replication was shown to be inhibited by NH_4_Cl treatment include alphaviruses [55], herpes simplex virus type 1 [56], reoviruses [57], and coronaviruses [58,59]. In the present study, NH_4_Cl treatment neither inhibited nor enhanced TiLV replication in E-11 cells, as there was no difference in virus replication in NH_4_Cl-treated or -nontreated cells.

Altogether, the data presented here show that TiLV does not agglutinate avian and piscine RBCs and its ability to replicate in NH_4_Cl-treated cells suggests that it could use uptake mechanisms that are different from orthomyxoviruses. Therefore, future studies should seek to elucidate the mechanisms by which TiLV binds to host cells and its mode of entry into infected cells.

## Figures and Tables

**Figure 1 viruses-11-01152-f001:**
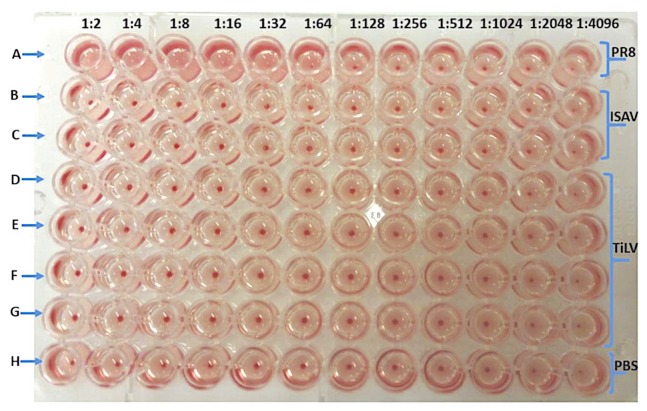
Hemagglutination of turkey red blood cells by PR8 (512HA/50 µL), ISAV (10^6^ TCID_50_/mL), TiLV (10^7^, 10^6^, 10^5^, and 10^4^ TCID_50_/mL in rows D, E, F, and G, respectively) after 1 h of incubation at room temperature.

**Figure 2 viruses-11-01152-f002:**
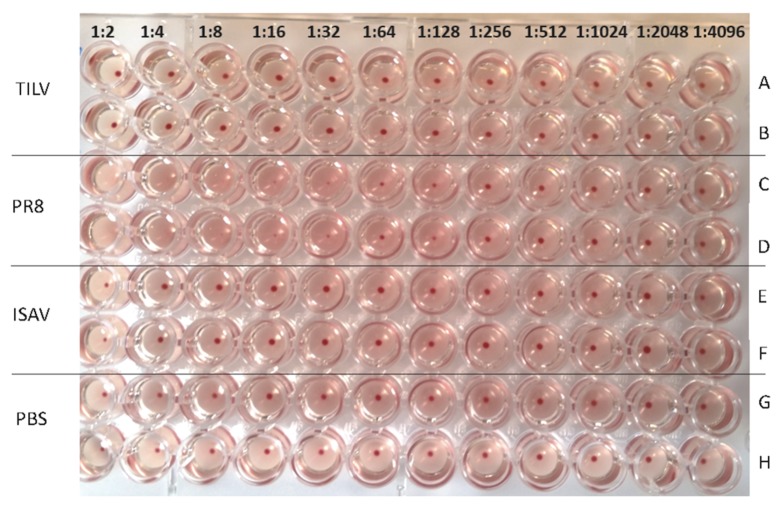
Hemagglutination of tilapia red blood cells by PR8 (512HA/50 μL), ISAV (10^6^ TCID_50_/mL), and TiLV (10^7^ TCID_50_/mL) after 1 h of incubation at room temperature.

**Figure 3 viruses-11-01152-f003:**
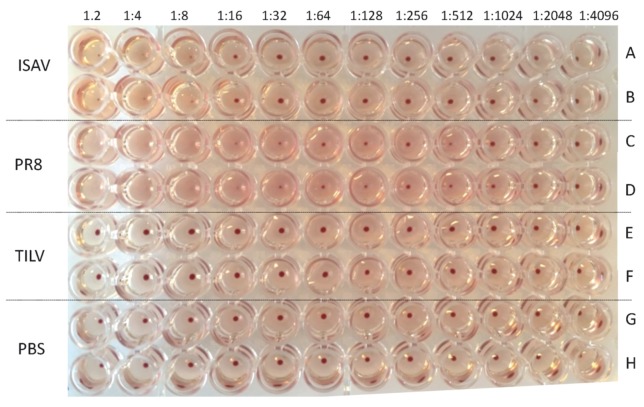
Hemagglutination assay of Atlantic salmon red blood cells by, ISAV (10^6^ TCID_50_/mL), PR8 (512HA/50 μL), and TiLV (10^7^ TCID_50_/mL) after 1 h of incubation at room temperature.

**Figure 4 viruses-11-01152-f004:**
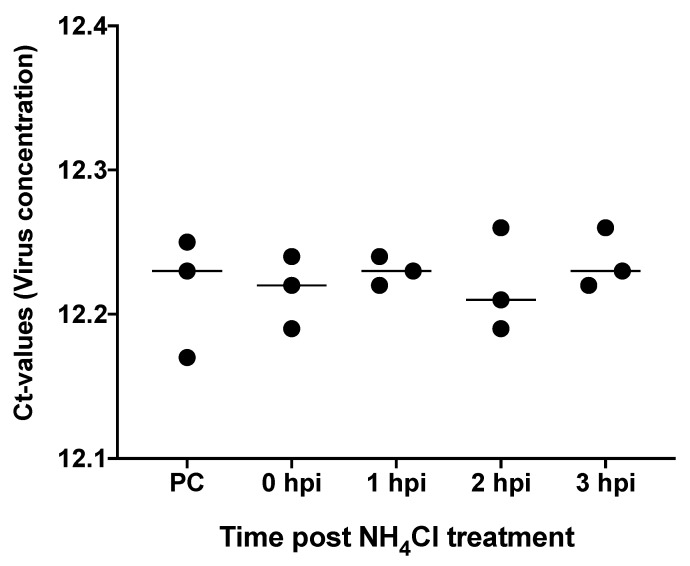
Mean viral loads of NH_4_Cl-treated E-11 cells infected with TiLV at 0, 1, 2, and 3 h. Quantification of tilapia lake virus (TiLV) in E-11 cells nontreated and treated with NH_4_Cl using quantitative real-time PCR (qRT-PCR). *C*t-values depict the quantity of viral RNA detected by qRT-PCR at 3 days post-infection (dpi) from three replicates of E-11 cells treated with NH_4_Cl at 0, 1, 2, and 3 h post-infection with TiLV.

**Table 1 viruses-11-01152-t001:** List of primers used for verifying the viruses (tilapia lake virus (TiLV), influenza A/Puerto Rico/8 (PR8), and infectious salmon anemia virus (ISAV)).

Primer Name	Primer Sequence	Expected Product (bp)	Virus/Segment	Reference
TiLV-SG3-F4	TCCAGATCACCCTTCCTACTT	109	TiLV/3	This study
TiLV-SG3-R4	ATCCCAAGCAATCGGCTAAT			
Flu A-F	TAACCGAGGTCGAAACGTA	195	PR8/7	[29]
Flu A-R	GCACGGTGAGCGTGAA			
H520-F	CTACACAGCAGGATGCAGATGT	104	ISAV/8	[30]
H534-R	CAGGATGCCGGAAGTCGAT			

**Table 2 viruses-11-01152-t002:** Hemagglutination of TiLV, PR8, and ISAV of avian and piscine red blood cells (RBC).

Viruses	Hemagglutination Titre (Dilution Factor/50 μL) of Avian and Piscine Erythrocytes		
	Turkey RBCs	Tilapia RBCs	Atlantic Salmon RBCs
PR8	1:128	1:64	1:512
ISAV	0	1:4	1:16
TiLV	0	0	0

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
