# Peer review of "Tilapia Lake Virus Does Not Hemagglutinate Avian and Piscine Erythrocytes and NH4Cl Does Not Inhibit Viral Replication In Vitro"

_viruses, 2019, doi:10.3390/v11121152_

Round 1

Reviewer 1 Report

The manuscript by Chengula et al. is an interesting,  well-written and well-executed report on studies conducted to investigate behavioural properties of a relatively new and unknown virus, with sound conclusions and suggestions for further investigations.

I only had two relatively minor comments for the authors to consider:

Introduction: As well as the known molecular properties of the virus, it would be interesting for the reader to know something about the disease/clinical presentation caused by infection with TiLV as well as the epidemiological significance of the virus/disease. How is the disease is controlled/monitored? Brief details could included.

Line 97: Why was this particular influenza viru strain used in the study and what was previously known about this virus?

Author Response

REVIEWER-1

QUERY-1: Introduction: As well as the known molecular properties of the virus, it would be interesting for the reader to know something about the disease/clinical presentation caused by infection with TiLV as well as the epidemiological significance of the virus/disease. How is the disease is controlled/monitored? Brief details could included.

Response: Brief details on the transmission, clinical manifestation and control have been included in manuscript (see lines 44-49)

QUERY-2: Line 97: Why was this particular influenza virus strain used in the study and what was previously known about this virus?

Response: We were not particular to this strain but rather a representative of all the influenza viruses as TiLV was suggested to be related with influenza viruses based on virion structure and genome segmentation. Moreover, this widely used for experimental use because the strain is less pathogenic to humans.

Reviewer 2 Report

Comments to viruses-641471

A review of the manuscript entitled Tilapia lake virus does not hemagglutinate avian and piscine erythrocytes and NH4Cl does not inhibit viral replication in vitro by Augustino Alfred Chengula, Stephen Mutoloki, Øystein Evensen and Hetron Mweemba Munang’andu.

The manuscript describes functional characterization of a novel orthomyxo-like virus first characterized in 2014 and causes disease in Tilapia. The work focuses on two methods for characterizing TiLV, hemagglutination assays and the effect of NH4Cl on viral replication. Other orthomyxoviruses such as influenza A, B and C viruses and infectious salmon anaemia virus (ISAV) have the ability, mediated by their surface glycoproteins HA, HEF, HE respectively, to agglutinate red blood cells from different species in vitro. NH4Cl inhibits replication of these viruses by interfering with the lowering of the pH in endosomal compartment following endocytosis, and hence cleavage of one of the surface glycoproteins necessary for fusion between the viral and endosomal membranes.

There are issues throughout the paper that must fixed. Many of these are indicated below. Both the text and figures in the results section would benefit from shortening or be omitted. The language and structure within sections also needs improvement throughout the manuscript. The plates showing the stripping (Fig. 2B, 3B and 4B) appear identical or marginally different to those shown in the Figs. 2-4A. The results from the Fig. 2-4B should be mentioned, but the photos could be put in supplementary material. I also do miss a comment or sentence specifying that HA titers used for PR8 have no direct links to the TCID50 data used for TiLV and ISAV. The former is an indicator of the amount of HA, the ladder is infectivity. Hence, different types of virus standardization methods are used for the hemagglutination assays. Also, there seems to be some mixup regarding HA titers and dilution factor some places.

In short, the manuscript does provide some new functional information on this important virus causing disease in Tilapia, information that could also aid in proper classification of this virus. But it is too long and has a number of issues, many outlined below.

Below issues that must be addressed or corrected:

Line 12: negative-sense single-stranded, -ssRNA, icosahedral virus (other places as well).

Line 35: Reference 1 is duplicated in reference list.

Line 40-42: Rephrase to …motif conserved in RNA-dependent RNA and RNA-dependent DNA polymerases…

Line 46: Hemagglutinin (HA) and neuraminidase (NA/NB) only in influenza A and B viruses. Change orthomyxoviruses to those.

Line 53: of conserved

Line 53: Influenza A and C viruses (with regards to the reference used)

Line 46: Not all orthomyxoviruses have HA and NA. ISAV has HE and F. INVC has HEF. Replace orthomyxoviruses with influenza A viruses.

Line 64: agglutinate

Line 70: delivery into cytoplasm

Line 70-74: sentence to long

Line 75: endosomal membrane inhibitors? Does it inhibit the membrane?

Line 78: orthomyxoviruses? Or only influenza viruses and ISAV?

Line 78-81: poor sentence with information missing; any species included in the present study, absence of surface proteins with hemagglutinating properties etc..

Line 133: working solution

Line 136-137: edit sentence

Line 147-150: poor sentence, rephrase. A reference were this has been used? 600C angle?

Line 186: The figure with the agarose gel could very well be put in supplementary material as it just represents a confirmation of the viruses. Mentioning how they were verified by PCR is adequate.

Line 187: ladder

Figure 1 might be considered put in supplementary material.

Line 191: …virus… PR8 HA/50 ul, is this correct. The strain is named PR8.

Line 196-197: poor sentence, are 128 HA titers being mixed up with the 128 dilution factor? Same question for Table 2.

Line 207-211: Figure 2 legend is repetitive. Also, I would not use the terms TiLV1-4 for the different concentrations. This could be mistaken for different isolates or subtypes.

Line 235-239: Figure 3 legend to repetitive. Also, it says both Tilapia and salmon RBCs in the legend text..?

Line 226: You haven’t specified HA units for ISAV, only TCID50 and the dilutions. Have you normalized it towards that of PR8? Look like highest dilution factor for ISAV agglutination 1:16 in Fig. 4. If you were to convert that dilution to HA units i.e. PR8, that would correspond to 32 units.

Line 225-233: Difficult to follow text with figure. Revise.

Line 227: Same issue as above. The dilution is 1:512, not 512 HA units.

Line 242-247: Figure 4 legend to repetitive and disorganized (same for those in figs 2 and 3). Also, both Tilapia and salmon RBCs in the legend text here also.

Lines 249- 271: The language has to be improved. And the whole section should be shortened significantly, it is way too long.

Figure 5 could also be put in supplementary material. Fig. 6 is sufficient to describe the results from NH4Cl treatment in the main manuscript.

Line 282: quantity of viral RNA not virus.

Line 289: I would say the agglutination of tilapia RBCs by ISAV is very weak.

Line 290: ..of the..

Line 298: …human influenza A and B viruses…

Author Response

REVIEWER-2

QUERY-1: There are issues throughout the paper that must fixed. Many of these are indicated below. Both the text and figures in the results section would benefit from shortening or be omitted. The language and structure within sections also needs improvement throughout the manuscript. The plates showing the stripping (Fig. 2B, 3B and 4B) appear identical or marginally different to those shown in the Figs. 2-4A. The results from the Fig. 2-4B should be mentioned, but the photos could be put in supplementary material. I also do miss a comment or sentence specifying that HA titers used for PR8 have no direct links to the TCID50 data used for TiLV and ISAV. The former is an indicator of the amount of HA, the ladder is infectivity. Hence, different types of virus standardization methods are used for the hemagglutination assays. Also, there seems to be some mixup regarding HA titers and dilution factor some places.

Response: The photos for stripping (Fig. 2-4B) have been moved in the supplementary material but mentioned in the text. We have stated in the manuscript in the result section that there was no direct link between the HA titers used for PR8 and the TCID50 data used for TiLV and ISAV. The mix up regarding HA titres and dilution factor has been corrected in the manuscript.

QUERY-2: Line 12: negative-sense single-stranded, -ssRNA, icosahedral virus (other places as well).

Response: corrected in the manuscript (see lines 12 and 37)

QUERY-3: Line 35: Reference 1 is duplicated in reference list.

Response: The duplicate reference 1 in the reference list has been removed (see reference list)

QUERY-4: Line 40-42: Rephrase to …motif conserved in RNA-dependent RNA and RNA-dependent DNA polymerases…

Response: Rephrased in the manuscript (see lines 40-41)

QUERY-5: Line 46: Hemagglutinin (HA) and neuraminidase (NA/NB) only in influenza A and B viruses. Change orthomyxoviruses to those.

Response: Has been changed to influenza A and B viruses in the manuscript (see line 51)

QUERY-6: Line 53: of conserved

Response: corrected in the manuscript (see line 58)

QUERY-7: Line 53: Influenza A and C viruses (with regards to the reference used)

Response: corrected in the manuscript (see line 58)

QUERY-8: Line 46: Not all orthomyxoviruses have HA and NA. ISAV has HE and F. INVC has HEF. Replace orthomyxoviruses with influenza A viruses.

Response: Replaced in the manuscript (see line 51)

QUERY-9: Line 64: agglutinate

Response: corrected in manuscript (see line 70)

QUERY-10: Line 70: delivery into cytoplasm

Response: corrected in the manuscript (see lines 74-75)

QUERY-11: Line 70-74: sentence to long

Response: The sentence has been divided into two (see lines 75-79)

QUERY-12: Line 75: endosomal membrane inhibitors? Does it inhibit the membrane?

Response: Endosomal membrane inhibitors has been replaced with endosomal membrane fusion inhibitors (See line 80)

QUERY-13: Line 78: orthomyxoviruses? Or only influenza viruses and ISAV?

Response: orthomyxoviruses replaced with influenza viruses and ISAV (see line 82)

QUERY-14: Line 78-81: poor sentence with information missing; any species included in the present study, absence of surface proteins with hemagglutinating properties etc..

Response: The sentence has been improved to incorporate the reviewer’s suggestion (see line 83)

QUERY-15: Line 133: working solution

Response: corrected in the manuscript (see line 138)

QUERY-16: Line 136-137: edit sentence

Response: The sentence has been edited in the manuscript (see lines 141-142)

QUERY-17: Line 147-150: poor sentence, rephrase. A reference where this has been used? 600C angle?

Response: The sentence has been rephrased has been used inserted in manuscript (see lines 152)

QUERY-18: Line 186: The figure with the agarose gel could very well be put in supplementary material as it just represents a confirmation of the viruses. Mentioning how they were verified by PCR is adequate.

Response: The figure (Fig.1) with agarose gel has been put in the supplementary material (See Figure S1, S1 implies supplementary 1)

QUERY-19: Line 187: ladder

Response: Corrected in the manuscript (see line 553)

QUERY-20: Figure 1 might be considered put in supplementary material.

Response: The figure (Fig.1) has been put in the supplementary material and is shown as Figure S1, See lines 550-551)

QUERY-21: Line 191: …virus… PR8 HA/50 ul, is this correct. The strain is named PR8.

Response: The HA/50 ul has been removed in the manuscript (see line 191)

QUERY-22: Line 196-197: poor sentence, are 128 HA titers being mixed up with the 128 dilution factor? Same question for Table 2.

Response: The sentence has been improved and the 128 HA titers changed to 1:128 dilution factor and changed accordingly in Table 2 (see line 197 and Table 2).

QUERY-23: Line 207-211: Figure 2 legend is repetitive. Also, I would not use the terms TiLV1-4 for the different concentrations. This could be mistaken for different isolates or subtypes.

Response: The repetition in the figure legend has been removed and the terms TiLV1-4 in figures have been changed to avoid confusion (see 208).

QUERY-24: Line 235-239: Figure 3 legend to repetitive. Also, it says both Tilapia and salmon RBCs in the legend text.?

Response: The repetition has been removed and the fish species corrected accordingly (see lines 236-237).

QUERY-25: Line 226: You haven’t specified HA units for ISAV, only TCID50 and the dilutions. Have you normalized it towards that of PR8? Look like highest dilution factor for ISAV agglutination 1:16 in Fig. 4. If you were to convert that dilution to HA units i.e. PR8, that would correspond to 32 units.

Response: We did not normalize the TCD50 for the ISAV and TiLV towards that of PR8. Therefore, HA titers used for PR8 have no direct links to the TCID50 data used for TiLV and ISAV. This is explained in the manuscript, see lines 193-194

QUERY-26: Line 225-233: Difficult to follow text with figure. Revise.

Response: The text has been revised in the manuscript, see 194-198

QUERY-27: Line 227: Same issue as above. The dilution is 1:512, not 512 HA units.

Response: We have changed in the manuscript, see concentration is used 512 HA units and not dilution 1:512 (see lines 192).

QUERY-28: Line 242-247: Figure 4 legend to repetitive and disorganized (same for those in figs 2 and 3). Also, both Tilapia and salmon RBCs in the legend text here also.

Response: The repetitions in the legends for all the figures have been removed and organized. Also, RBCs for respective species for each legend has been corrected (see line 184).

QUERY-29: Lines 249- 271: The language has to be improved. And the whole section should be shortened significantly, it is way too long.

Response: The whole section has been shortened and the language improved (see line 196-198 and Section 190-204)).

QUERY-30: Figure 5 could also be put in supplementary material. Fig. 6 is sufficient to describe the results from NH4Cl treatment in the main manuscript.

Response: Figure 5 has been put in the supplementary material (see lines 570-571, Figure S5).

QUERY-31: Line 282: quantity of viral RNA not virus.

Response: Changed in the manuscript, See line 259

QUERY-32: Line 289: I would say the agglutination of tilapia RBCs by ISAV is very weak.

Response: We have incorporated the suggestion in the manuscript (see line 267)

QUERY-33: Line 290: ..of the..

Response: corrected in the manuscript (see line 267)

QUERY-34: Line 298: …human influenza A and B viruses…

Response: corrected in the manuscript, see lines 275-276

Round 2

Reviewer 2 Report

Comments to revised viruses-641471

The language in the results section could still be improved somewhat.

Table 2 should be placed after Fig. 3 as it summarizes the hemagglutination results.

Line 184: Figure S1 in parenthesis.

Line 185-186: ..was in concordance with the expected amplicon product sizes for each virus using the primers shown in Table 1.

Line 186: RNase

Lines 186-188: poor sentence, rephrase.

Line 219: evident

Line 225-226: Row D and E, Fig 4A and 4B?

Line 250: a typo, remove “s”

Line 253: Ct values obtained by

Figure 3 legends: skip “assay”

Figure S2 & S3 legend: ..influenza virus PR8..